# Structural Concept Learning via Graph Attention for Multi-Level Rearrangement Planning

**Manav Kulshrestha, Ahmed H. Qureshi**
Department of Computer Science, Purdue University
West Lafayette, IN 47907, United States
{mkulshre, ahqureshi}@purdue.edu

**Abstract:** Robotic manipulation tasks, such as object rearrangement, play a crucial role in enabling robots to interact with complex and arbitrary environments. Existing work focuses primarily on single-level rearrangement planning and, even if multiple levels exist, dependency relations among substructures are geometrically simpler, like tower stacking. We propose Structural Concept Learning (SCL), a deep learning approach that leverages graph attention networks to perform multi-level object rearrangement planning for scenes with structural dependency hierarchies. It is trained on a self-generated simulation data set with intuitive structures, works for unseen scenes with an arbitrary number of objects and higher complexity of structures, infers independent substructures to allow for task parallelization over multiple manipulators, and generalizes to the real world. We compare our method with a range of classical and model-based baselines to show that our method leverages its scene understanding to achieve better performance, flexibility, and efficiency. The dataset, demonstration videos, supplementary details, and code implementation are available at: https://manavkulshrestha.github.io/scl.

**Keywords:** Rearrangement Planning, Robot Manipulation, Graph Attention

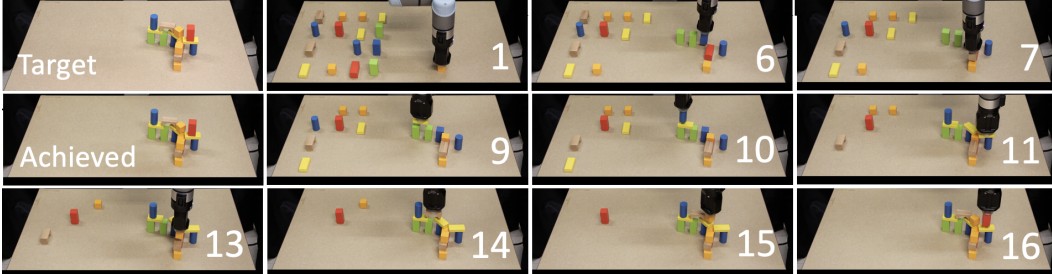

Figure 1: Our approach performing progressive pick-and-place (we only show some place steps) actions based on its multi-level rearrangement plan to achieve (middle left) a target arrangement based on a given goal (top left). A complete figure is available in the Appendix.

## 1 Introduction

Robots operating in the real world will often encounter a variety of simple objects in more complex arrangements and structures. To that end, recent years have seen rearrangement planning – which involves the reorganization of objects in a given environment to bring them into a goal state – emerging as a prominent area of research within robotics [1]. Simpler applications of this problem include tasks such as setting the table, rearranging furniture, loading a dishwasher, and many more. While manipulation for many real-world tasks performed by robots often reduces to pick-and-place, achieving more structured goals requires compositional interpretation of the target and the execution of long-horizon hierarchical plans. Autonomously solving more complex rearrangement problems,

7th Conference on Robot Learning (CoRL 2023), Atlanta, USA.

such as constructing a house or mechanical assembly, require robots to interpret internal dependence among the different parts of the whole structure and plan accordingly.

Despite the importance of these problems, most solutions addressing rearrangement planning largely consider target configurations with simpler dependencies such as blocking obstacles [2, 3, 4], stacking objects in towers or very simple structures [5, 6, 7, 8], or formulated as bin placement where supporting object is stationary [6, 9]; all of which are strict subsets of a more general construction task. One of the major hurdles for this task is to decipher the order in which objects need to be placed so as to properly construct a given target structure since some objects depend on others in that they geometrically support them.

In this paper, we explore a toy scenario of the construction problem with known object primitives which are to be arranged to create some target structure. Our approach takes a multi-view RGB-D observation of the target structure and constructs a dependency graph using graph attention networks. This graph captures the geometrical dependency among the different objects, identifying independent substructures for parallelization. Our structure planner then takes this graph and serializes task executions based on the current observations of the scene. The low-level controller further takes the task sequences and executes the tasks via robot control. Our results show better performance compared to other classical and model-based planners as well as generalizability to unseen structures of varying complexity. The main contributions of our approach are as follows:

- A generated data set and generation procedure of target scenes containing intuitive structures with diverse complexity due to the possibility of a variable number of objects, inclusion of structures with varying levels, and the possibility of independent substructures.
- A scalable dependency graph generator that generalizes to structures with multiple levels and varying numbers of objects.
- A structured planner which uses the dependency graph to create a sequential plan for parallelized multi-level rearrangement planning, which is integrated into a complete pipeline from scene observations to control executions.

## 2    Related Work

**Rearrangement.** The most relevant area of research to our work would be that of rearrangement planning, which is a subset of task and motion planning (TAMP). TAMP [10] involves planning for a robotic agent to operate in an environment with several objects by taking actions to move and change the state of said objects. Rearrangement – specifically proposed as an important challenge for embodied AI – narrows this by defining itself as the act of bringing a given environment to a goal state [1]. Various strategies are utilized by systems aiming to address these problems. Some methods propose the combination of a high-level task planner with a low-level motion planner [11, 12, 13, 14]. Others tackle the problem by using sampling-based techniques in conjunction with search algorithms [15, 16, 17]. However, all of these focus on a yet narrower subset of rearrangement where objects are largely restricted to a 2D workspace and the interplay between objects is largely ignored or not present beyond dealing with clutter.

**Deep Learning.** More recently, advances have been made which utilize deep learning-based approaches toward solving these problems. These allow relaxing the assumptions on objects involved by proposing more flexible collision detection [18, 9], better generalization to the real world [19, 20, 21, 4, 6, 2], and unseen environments [7, 2] through vision-based perception. Other methods utilize semantic information but use it to allow for more general goals like similarity to an inferred target distribution [22, 23, 24] or guidance using language [25, 26]. Perhaps most similar to our method, some approaches make use of graph neural networks to model object relations in the scene [8, 5]. However, in all the methods mentioned above, the target scenes are either restricted to a single-level or are much simpler in terms of their construction, and the spatially weaker relations allow for less closely dependent substructures.

**Construction and Assembly.** Some works explore the task of assembly or construction [27, 28, 29], which allows objects to become fixed to one another that, unlike our case, does not require the creation of stable non-collapsing structures. Furthermore, in contrast to our approach, Blocks

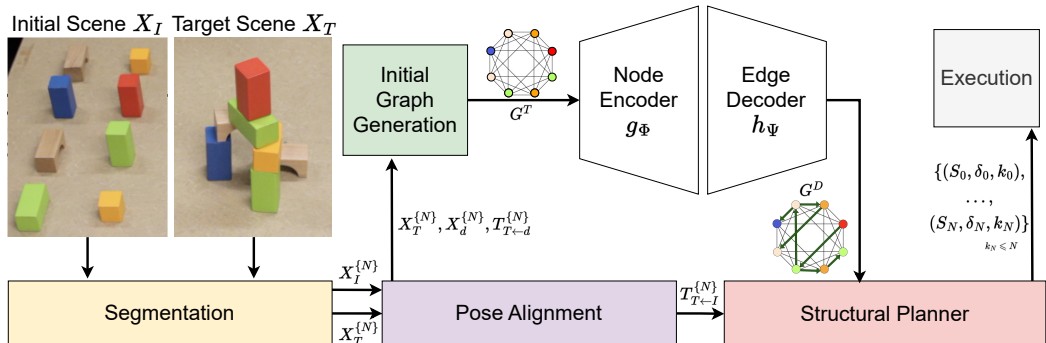

Figure 2: Model architecture overview. We segment out object point clouds (PointNet++ [30] based model), perform pose alignment, and establish object correspondences (TEASER++ [31] based) between $X_I$ and $X_T$. Object level embeddings (another PointNet++ [30] based model) and positional embeddings (positional encoder [32]) create an initial graph $G^T$ which our node encoder $g_\Phi$ uses to output higher-level node features. Our edge decoder $h_\Psi$ uses these to create a dependency graph $G^D$ from which the planner outputs a valid sequence for robot rearrangement.

[27] and RoboAssembly [28] also assume perfect information of objects in simulation, making generalization to the real world very difficult and requiring much higher planning steps due to the use of reinforcement learning. Additionally, while the aforementioned approach to construction [29] also performs long-horizon planning like ours, they restrict their target be one of 4 possible predefined structures, though their focus is on multi-robot planning.

## 3 Problem Definition

Let $\mathcal{X} = \{x_1, x_2, \ldots, x_n\} \subseteq \mathbb{R}^{n \times 3}$ be the set of all points that can be occupied by a scene, $\Xi$ be the powerset of $\mathcal{X}$, and $\mathcal{O} = \{o_1, o_2, \ldots, o_m\}$ be the set of all object instances in any given scene where each $o \in \mathbb{R}^6 \times \mathcal{C}$ with $\mathcal{C}$ being the set of all classes an object instance can take. Next, we define a scene as two sets of points $X, Y \in \Xi$ such that $X \subseteq Y$ where $X$ represents the observable scene, whereas $Y$ represents the complete scene. Furthermore, we can define a subset selection operator as $S \in \mathcal{S}$ as $S : \Xi \mapsto \Xi$ where $S(X) \subseteq X$ for all $X \in \Xi$. The goal is to construct a high-level planner $\pi_H : \Xi \times \Xi \mapsto \mathcal{P}$ that acts on partially observable sets $X_I, X_T \in \Xi$ of the initial and target scene to produce a hierarchical plan $P = \{(S_0, \delta_0), \ldots, (S_M, \delta_M)\} \in \mathcal{P}$ where $S_i \in \mathcal{S}$ is a subset selection operator, $\delta_i \in \Delta \subseteq \mathrm{SE}(3)$ is a valid spatial transformation, $M \in \mathbb{N}$ indicates the number of steps, and each $i^{\text{th}}$ step $p_i = (S_i, \delta_i) \in P$ contains the selection and transformation action. Hence, our objective is to determine a plan $P$ that selects subsets of point clouds, $\{S_0(X_I), \cdots, S_M(X_I)\}$, in the initial scene, $X_I$, and transform them using $\{\delta_0, \cdots, \delta_M\}$, in the minimum number of steps, $M$, to reach the an achieved state $X_A$ whose object specific point clouds are subsets of those given by the complete target state $Y_T$. For application to robot rearrangement, we further define a low-level planner $\pi_L : \mathcal{Q} \mapsto \mathcal{A}$ where $\mathcal{Q}$ and $\mathcal{A}$ refer to the configuration and action spaces for the robot. Every step $p_i = (S_i, \delta_i) \in P$ from the plan output by $\pi_H$ will have a sequence of achievable configurations $q_{\delta_i}$ associated with it that the low-level planner will take and further produce a sequence of actions $A \in \mathcal{A}$ to achieve the intermediate state defined by the application of $p_i$ on the previous state. For convenience and brevity, we will denote some more notation for the remainder of this paper. Let any arbitrary set $u$, where $|u| = n$, be denoted as $u^{\{n\}}$. Also, for any arbitrary set $U$, we denote its association with a scene $\iota$ by specifying a subscript as $U_\iota$ and a subset of $U$ associated some object or characteristic $\omega$ as a superscript $U^\omega \subseteq U$. And, for any graphs, superscripts denote different graph instances: $G^T, G^Z, G^D$.

## 4 Method

This section details our approach for multi-level rearrangement planning to generate a multi-step plan, the execution of which results in achieving the unseen structured target scene. Figure 2 shows an overview of the model architecture and the basic flow of the approach. Additional model and implementation details are available in Section 7.2 of the Appendix.

**Point Cloud and Feature Extraction.** Partial point clouds are generated from the target scene and initial scene images. First, we obtain the RGB-D images from multiple viewpoint cameras surrounding the scene area and calculate their corresponding point clouds in the world frame from the camera's known world frame positions. We do preliminary filtering to remove outlier points. This is done for both the target scene and the initial scene to obtain $X_T$ and $X_I$. A trained PointNet++[30] based segmentation network then takes the scene point cloud and returns the segmented identity values for each point. Next, we extract the object-specific point clouds to obtain $X_T^o$ and $X_I^o$ for each object $o$. This makes up the Segmentation Module. As part of the Initial Graph Generation, we use another PointNet++ based network to extract object-level latent features for each object point cloud and concatenate them to get $w_I^{\{N\}}$ and $w_T^{\{N\}}$ which will act as part of the node features for the scene graphs. Note that $N$ is the number of objects in each scene.

**Object Pose Alignment.** Now is the task of object alignment and correspondence creation. Between the target and initial scenes, we have multiple objects of the same type so we want to select correspondences such that the orientation change for an object $o_i$ is minimized from the initial to the target scene. To that end, for each object $X_T^{o_i}$ in the target scene, we use its predicted $y_T^{o_i}$ identity and sample its known mesh to obtain a default surface point cloud $X_d^{o_i}$. These complete, $X_d^{o_i}$, and the partial point clouds, $X_T^{o_i}$, from the target scene, are then used to predict a spatial transformation $\mathcal{T}_{T \leftarrow d}^{o_i}$, using TEASER++ [31], which aligns $X_d^{o_i}$ with $X_T^{o_i}$. This process is repeated for each object point cloud $X_I^{o_j}$ in the initial scene matching whose identity $y_I^{o_j}$ matches $o_i$'s identity $y_T^{o_i}$ to get a set of transformations $\{\mathcal{T}_{d \leftarrow I}^{o_0}, \mathcal{T}_{d \leftarrow I}^{o_1}, \mathcal{T}_{d \leftarrow I}^{o_2} \ldots\}$. These are then right multiply with $\mathcal{T}_{T \leftarrow d}^{o_i}$ to obtain $\{\mathcal{T}_{T \leftarrow I}^{o_0}, \mathcal{T}_{T \leftarrow I}^{o_1}, \mathcal{T}_{T \leftarrow I}^{o_2} \ldots\}$. From these, we choose the one which minimizes the magnitude of rotation from the initial to target scene (to avoid reorienting with multiple pick-and-place actions). This gives us the canonical correspondence (and associated transformations $\mathcal{T}_{T \leftarrow I}^{o_i}, \mathcal{T}_{T \leftarrow d}^{o_i}$) for $o_i$ between the initial and target scene. This makes up the Pose Alignment Module. Finally, as part of the Initial Graph Generation, we obtain centroids for $X_d^{o_i} \cdot (\mathcal{T}_{T \leftarrow d}^{o_i})^T$ and $X_d^{o_i} \cdot (\mathcal{T}_{I \leftarrow d}^{o_i})^T$ which are put into a positional encoder [32] to obtain positional features for each object in both the initial and target scenes: $b_I^{\{N\}}, b_T^{\{N\}}$. Together, the object level features and positional features give us the node features $n_I^{\{N\}} = [w_I^{\{N\}} \,||\, b_I^{\{N\}}], n_T^{\{N\}} = [w_T^{\{N\}} \,||\, b_T^{\{N\}}]$ for the objects in the scenes for the initial graph, where $||$ denotes concatenation.

**Graph Node Encoder.** The graph node encoder, $g_\Phi : \mathcal{G} \mapsto \mathcal{Z}$ is based on a graph attention neural network [33, 34], known as GAT, which takes in an initially fully connected scene graph $G^T = (V, E) \in \mathcal{G}$ of the target scene with the aforementioned $n_T^{\{N\}}$ – which contain both object-level features and positional features for each object in the target scene – serving as the initial node features. These initial node features are updated using a modified convolution that occurs for any node $i$ using its neighbor set $\mathcal{N}(i)$. The exact node feature update done by the convolutional layer is given by $n_i' = \alpha_{i,i}\Phi n_i + \sum_{j \in \mathcal{N}(i)} \alpha_{i,j}\Phi n_i$, where $\Phi$ is the learnable parameter for the update mechanism and the attention coefficients $\alpha$, which quantifies the importance of neighboring nodes, is given by

$$\alpha_{i,j} = \frac{\exp(a^T \text{LeakyReLU}(\Phi[n_i \,||\, n_j]))}{\sum_{k \in \mathcal{N}(i) \cup \mathcal{N}(j)} \exp(a^T \text{LeakyReLU}(\Phi[n_i \,||\, n_k]))} \tag{1}$$

where $^T$ represents transposition and $a$ is the learnable parameter for the attention mechanism (for derivation and more details, please refer to [33, 34]). The output from the graph encoder is a latent scene graph $G^Z \in \mathcal{Z}$ for the target scene containing high-level features for each of the objects. In our problem setting, a GAT-based model with multi-headed attentions being averaged outperformed a vanilla GCNs.

**MLP Edge Decoder.** The MLP-based edge decoder $h_\Psi : \mathcal{Z} \to \mathcal{D}$ maps the latent scene graph $G^Z$ to the structural dependency graph $G^D$ for the target scene containing inter-object specific dependency information. Specifically, $h_\Psi$ can be queried with a pair of high-level features $z_i, z_j$ representing objects $o_i, o_j$ and will decode them into the structural relationship between them. This relationship of dependence is asymmetric, and the decoder is used to query every ordered pair of nodes in the graph to obtain the respective dependence probabilities $\rho^{\{N \times N\}}$ where $\rho_{i,j} = h([z_i||z_j]; \Psi)$. Given

$\rho^{\{N \times N\}}$, we construct an inferred adjacency matrix for the structural dependency graph $G^D$ of the target scene with values $G_{i,j}^D = \rho_{i,j} > t^*$ where $t^*$ is some threshold value. The directed edges for this graph are visualized in Figure 2 as green arrows.

**Structured Plan Creation.** Once we have a directed acyclic graph $G^D$ representing the inferred structural relationship between each pair of objects different objects in the target scene, we perform a topological sorting to obtain a valid sequencing with which the objects can be introduced so as to construct the structure in the target scene. Next, we make use of the aforementioned object correspondences between the initial and target scene to pick the object from its current position (formulated as a point cloud selection of $S$) in the initial scene and place it in the target position given by the associated spatial transformation $\delta = \mathcal{T}$ obtained from pose alignment. All of this provides us with a plan $P = \{(S_0, \delta_0, k_0), \dots, (S_N, \delta_N, k_N)\}$ where $N$ is the number of objects. For plan step $p_i = (S_i, \delta_i, k_i) \in P$, $k_i \leqslant N \in \mathbb{N}$ denotes the dependence hierarchy identifier in the target structure that the object represented by the selection $S_i$ belongs to. Particularly, the plan is agnostic to any inversions involving any plan steps $p_i, p_j \in P$ such that their associated hierarchy identifiers match: $k_i = k_j$. That is, objects belonging to the same class of dependence hierarchy can be placed in any order relative to each other. This makes up the Structural Planner Module.

**Planning Algorithm.** Algorithm 1 defines our planning algorithm, which yields a multi-step plan for performing efficient multi-level rearrangement. For given scenes, we extract the observable information and convert them into scene point clouds $X_I, X_T$. These are taken and segmented into a collection of object point clouds $X_I^{\{N\}}, X_T^{\{N\}}$ and their respective predictive identities $y_I^{\{N\}}, y_T^{\{N\}}$, where $N$ is the number of objects in our scenes (Lines 1-2). Using these, we calculate the correspondences for objects in the initial and target scene to produce reordered versions of the object point cloud along with transformations $\mathcal{T}_{T \leftarrow I}^{\{N\}}$ for each object from the initial to target scene and transformations $\mathcal{T}_{T \leftarrow d}^{\{N\}}$ from objects' known default point cloud to their respective counterparts in the target scene (Line 3). Next, we use the target object point clouds $X_T^{\{N\}}$ with a PointNet++ [30] based model to extract object-level features $w_T^{\{N\}}$ and the default point cloud transformed to the target pose with a positional encoder [32] to get higher level embeddings $b_T^{\{N\}}$ for the target position (Lines 4-6). Together these make up and construct the initial graph $G^T$, which is provided to the graph encoder network $g_\Phi$ to get a graph with higher level node features $z^{\{N\}}$ for each object $G^Z$ (Line 7). Using these, we do ordered pairwise queries to the edge decoder $h_\Psi$ to populate the probabilities $\rho^{\{N \times N\}}$ of the $i^{\text{th}}$ object depending on the $j^{\text{th}}$ object which are used along with a threshold $t^*$ to determine the canonical structural dependency graph $G^D$ for the target scene (Lines 8-11). Given $G^D$, we now do a preliminary check of the inferred dependence by seeing whether the dependence graph is directed acyclic because if this is not the case, a circular dependency exists. Since all target scenes are known to be reachable with one manipulator, a cycle indicates a prediction error and we report failure (Lines 12-13). Finally, we do a topological sorting of the dependence graph $G^D$ and iterate over the result, using the predicted object transformations $\mathcal{T}_{T \leftarrow I}^{\{N\}}$ from the initial scene to the target to create a rearrangement plan $P$ (Lines 14-18). For each element $p_i = (S_i, \delta_i, k_i) \in P$, the execution will pick the object represented by the set of points $S_i(X_I)$ and execute low level actions to result in an effective transformation of $\delta_i = \mathcal{T}_i$ on said points in an order such that $k_i$ increases monotonically.

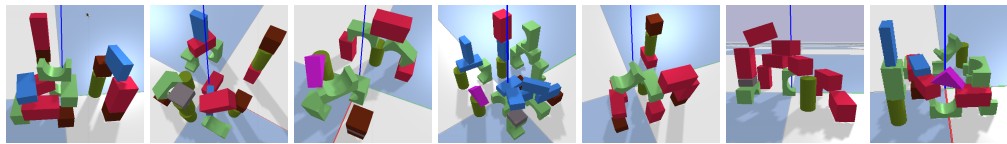

Figure 3: Examples of generated structures in dataset. For scale, the red cuboid is 3x3x6 cm$^3$

**Data Generation and Training.** To train our models, we generate synthetic data containing intuitive structures built from a set of 8 possible object primitives, which are captured from 3 fixed viewpoints. We restrict the set of initial orientations for each object to allow only those which provide a non-negligible surface area on the bottom surface to allow for stable placement (e.g., a

**Algorithm 1:** SCL-Planning($\mathcal{X}$)

---

**1** $X_I, X_T \leftarrow$ PointCloudExtraction($\mathcal{X}$)     ▷ initial and target scene point-clouds (pcds)

**2** $X_I^{\{N\}}, X_T^{\{N\}}, y_I^{\{N\}}, y_T^{\{N\}} \leftarrow$ Segmentation($X_I, X_T$)     ▷ object pcds and their identities

**3** $X_{I,T,d}^{\{N\}}, \mathcal{T}_{T \leftarrow I}^{\{N\}}, \mathcal{T}_{T \leftarrow d}^{\{N\}} \leftarrow$ PoseAlignment($X_I^{\{N\}}, X_T^{\{N\}}, y_I^{\{N\}}, y_T^{\{N\}}$) ▷ align and correspond

**4** $w_T^{\{N\}} \leftarrow$ ObjectFeatures($X_T^{\{N\}}$)

**5** $b_T^{\{N\}} \leftarrow$ PositionalFeatures($X_d^{\{N\}}, \mathcal{T}_{T \leftarrow d}^{\{N\}}$)

**6** $n_T^{\{N\}} \leftarrow$ NodeFeatures($w_T^{\{N\}}, b_T^{\{N\}}$)     ▷ node features for initial graph

**7** $G^Z = (z^{\{N\}}, E) \leftarrow g_\Phi(G^T = (n_T^{\{N\}}, E))$     ▷ graph with higher level node features

**8** $\rho^{\{N \times N\}} \leftarrow \emptyset$

**9 for** $(i, j) \in E$ **do**

**10**     $\rho_{i,j}^{\{N \times N\}} = h_\Psi([z_i^{\{N\}} \| z_j^{\{N\}}])$     ▷ edge decoding for each pairwise edge

**11** $G^D = (N, \rho^{\{N\}} > t^*)$     ▷ creation of dependency graph based on threshold

**12 if** IsNotDAG($G^D$) **then**

**13**     **return** Circular Dependency Failure     ▷ detecting unrecoverable prediction error

**14** $P^{\{N\}} \leftarrow \emptyset$

**15 for** $i, k_i \in$ TopologicalSorting($G^D$) **do**

**16**     $S_i, \delta_i \leftarrow$ RearrangmentStep($i, \mathcal{T}_{T \leftarrow I}^{\{N\}}$) ▷ $i^{\text{th}}$ object selection operator and transformation

**17**     $P \leftarrow P \cup \{(S_i, \delta_i, k_i)\}$     ▷ adding step $i$ to plan

**18 return** $P$

---

cylinder is not allowed to be placed in a way where it may roll). And, for all but the final layer, only object orientations that provide a stable surface of non-negligible surface area will be selected for placements (e.g., a pyramid will not be placed upright but may be placed sideways). For the $1^{\text{st}}$ layer, objects are initially placed randomly within the bounds of the target scene, ensuring no collision. Following this, a 2 step process is repeated for each $i^{\text{th}}$ layer. First, attempt to place objects supported by 2 objects in the previous layer. For this, we consider each pair of objects in the previous layer with available surface area and attempt to place an object in some valid orientation given the distance between the supporting objects. This is done by sampling some points on the top surface of the supporting objects and fitting a plane to them, which the initial valid orientation is projected onto to get the object pose. Note that this allows for random placements that are not constrained to be axis aligned. Once all such supporting pairs in the previous layer have been exhausted, we move on to the second phase. In the second phase, we repeat the process but attempt to place objects on top of just one object from the previous layer. This is repeated until the step budget is exhausted. In simulation, we calculate the ground truth dependency graph using the $y$-components of the contact force vectors between each pair of objects. The graph encoder $g_\Phi$ and edge decoder $h_\Psi$ were trained together in a supervised manner to minimize the binary cross entropy loss between the adjacency matrices of the predicted and ground truth dependency graphs. Some examples of generated structures are given in Figure 3. We use PyByllet [35] for the simulation environment and use Trimesh [36] for collision and geometric checking. More information, including the generation algorithm's pseudo-code is provided in Section 7.1 of the Appendix.

## 5 Results

We perform four sets of experiments. First, we tested our method on unseen structures in the simulated environment and compared it with some model-based and classical baselines. Second, we show how our method generalizes to multi-level object rearrangement tasks with structures containing a higher number of objects. Third, we show our method's generalization to multi-level object rearrangement tasks with unseen structures consisting of higher number of levels than were in the training set. Finally, we demonstrate our method's sim-to-real generalization on multi-level object rearrangement tasks in the real world. The real world experiments also show our method generalizing to and operating on structures with objects placed in locations that cross between different

| Planner | Performance Metrics | | | | |
|---|---|---|---|---|---|
| | Success (%) ↑ | Completion (%) ↑ | Steps ↓ | Pos Error (m) ↓ | Orn Error (0-1) ↓ |
| SCL (Ours) | 95.1 | 98.7 | $1.0 \pm 0.0$ | $0.002 \pm 0.003$ | $0.0024 \pm 0.0028$ |
| MLP | 76.2 | 81.7 | $1.0 \pm 0.0$ | $0.002 \pm 0.004$ | $0.0024 \pm 0.0031$ |
| Classical Random | 90.1 | 98.0 | $1.48 \pm 0.22$ | $0.002 \pm 0.004$ | $0.0029 \pm 0.0033$ |
| Classical Iterative | 45.8 | 81.8 | $1.56 \pm 0.25$ | $0.002 \pm 0.004$ | $0.0021 \pm 0.0025$ |

Table 1: Comparison between our approach with classical and model-based baselines. Our approach shows better-performing plans with fewer steps. Classic baselines were given a planning budget of $2N$ and take, on average, around 50% more steps for success than our approach, as indicated by the step factor.

levels – something also not present in the training data. Also note that the errors and steps were only calculated for successful cases.

**Evaluation Metrics.** We use the following metrics for quantitative comparisons of different tasks. *Success Rate:* The percentage of successfully solved unseen scenes where success is defined as no object's achieved pose differing from the target position by more than 1 cm or the target orientation by 0.03 (normalized quaternion distance). *Completion Rate:* The percentage of objects whose achieved pose was within the success threshold (defined above) of their target in the scene, averaged over all scenes. *Planning Steps:* The ratio of the planning steps required to rearrange the objects from the initial scene to the target scene and the number of objects in the scene, averaged over all scenes. *Position Error:* The mean Euclidean distance between an object's achieved and target position for all objects in the scene, averaged over all scenes. *Orientation Error:* The mean quaternion distance $\varphi(q_1, q_2) = \min\{||q_1 + q_2||_2, ||q_1 - q_2||_2\}$ [37], normalized to be between 0 and 1, between the quaternions representing the achieved and the target orientations for all objects in the scene, averaged over all scenes.

**Baselines.** The following baseline planners take over once the object correspondences between the target and initial scenes have been calculated. *Classical Iterative Baseline:* This is a model-free planner which iteratively selects objects from the initial scene that are not yet in their target pose. Once it has a candidate selected, it checks whether they can be stably placed in their target pose without falling by attempting a set of ray checks on the target location. If the check fails, it continues the iteration. Once it finds a valid object for placement, it places the object and restarts its iteration. *Classical Random Baseline:* This is a model-free planner which randomly selects objects from the initial scene that are not yet in their target pose. Once it selects a candidate, it does a stability check similarly to the other classical planner. If the check fails, it removes said object from its selection pool and continues random selection. If the check succeeds, it moves the object to its target pose and resets its selection pool to the current set of objects not in its target pose. *MLP Baseline:* This is an MLP-based object selection network that was trained on node features $[n_i^{\{N\}} \,||\, n_T^{\{N\}}]$ obtained from the scene after the execution of step $p_i$ from a ground truth plan with the goal of having the selection network learn and output the correct object selection operation $M_i \in p_i$ for planning. $N$ was set to a maximum value of 10 and node features were padded with zeros if the scene had fewer objects.

**Comparison Analysis.** Table 1 shows our comparison results with the aforementioned baselines, evaluated on upwards of 400 scenes of unseen structures. Our approach outperforms all baselines in terms of success and completion rate. The MLP baseline had a tendency to get stuck in a local minimum for which object to move next, causing it to sometimes not be able to attempt rearrangement of the remaining objects, resulting in a lower completion rate. Classical baselines were given a budget of $2N$ steps (where $N$ is the number of objects in a scene) and, on average, take around 50% more steps for success. Our approach generalizes to multiple objects, unlike MLP, and provides better performance in fewer steps compared to the classical approaches.

**Scalability Analysis.** Table 2 shows our method's ability to generalize to scenes with a variable number of objects, evaluated over more than 400 unseen scenes of structures with at most 3 layers. Our approach performs very well in scenes with less than 15 objects despite having been trained using a set containing an average of 9 objects. While a higher number of objects in a scene cause

| Number of Objects ($N$) | Performance Metrics | | | |
|---|---|---|---|---|
| | Success (%) ↑ | Completion (%) ↑ | Position Error (m) ↓ | Orientation Error (0-1) ↓ |
| $8 \leqslant N \leqslant 10$ | 95.1 | 98.7 | $0.002 \pm 0.003$ | $0.0021 \pm 0.0028$ |
| $10 \leqslant N < 15$ | 93.9 | 98.2 | $0.002 \pm 0.003$ | $0.0024 \pm 0.0036$ |
| $15 \leqslant N < 20$ | 81.1 | 97.9 | $0.003 \pm 0.004$ | $0.0028 \pm 0.0043$ |
| $20 \leqslant N < 25$ | 73.7 | 96.7 | $0.003 \pm 0.004$ | $0.0031 \pm 0.0054$ |

Table 2: Performance of our approach as the number of objects in the scene increases. Larger number of objects in the same space results in occlusion, causing lower success rates but with the completion rates remaining high.

occlusion resulting in poorer prediction, our algorithm can still recover and deliver high completion rates even when success rates reduce. Table 3 shows the performance of our approach on over 1000 scenes containing meaningful structures with a variety of levels despite having been trained only on scenes containing at most 3 levels, showing its ability to generalize to novel and unseen structure configurations. On structures with less than 3 levels, it has a 100% success rate. While occlusion due to denser structures causes a lower success rate as the number of levels increases, our algorithm still maintains a high completion rate.

| Number of Levels | Performance Metrics | | | |
|---|---|---|---|---|
| | Success (%) ↑ | Completion (%) ↑ | Position Error (m) ↓ | Orientation Error (0-1) ↓ |
| 1 | 100.0 | 100.0 | $0.001 \pm 0.000$ | $0.0011 \pm 0.0004$ |
| 2 | 100.0 | 100.0 | $0.002 \pm 0.003$ | $0.0025 \pm 0.0038$ |
| 3 | 94.5 | 98.6 | $0.002 \pm 0.003$ | $0.0036 \pm 0.0045$ |
| 4 | 88.0 | 98.4 | $0.002 \pm 0.003$ | $0.0036 \pm 0.0038$ |
| 5 | 79.8 | 97.0 | $0.002 \pm 0.003$ | $0.0038 \pm 0.0039$ |

Table 3: Generalization of our approach to structures with higher levels. Meaningful structures with higher levels again cause occlusion, and we see a similar trend of lower success rates but good completion rates. Note that our method only trained on structures containing at most 3 levels.

**Sim2Real Generalization.** We performed a set of real-world experiments using a UR5e robot with a suction gripper and three Intel RealSense cameras for scene observation. We set up target structures ranging from 8 to 16 blocks with variable levels within the structure and block locations that cross between levels (example shown in Figure 1). Despite being trained in simulation with structures limited to 3 levels and discrete level layers, our method generalizes well to novel problem settings in the real world. The demonstration videos are provided on the project web-page.

# 6 Conclusions, Limitations, and Future Works

We presented Structured Concept Learning (SCL), a graph attention network-based approach for multi-level rearrangement planning. SCL is an end-to-end approach which infers the dependency of objects and substructures to build multi-level structures from point cloud sets of the initial and target scene from RGB-D cameras. Our approach was trained on a novel data set gathered by our intuitive multi-level structure generation procedure. In addition to demonstrating its sim-to-real generalization, we evaluate our approach on challenging problems defined by target scenes containing different unseen structures with a variety of objects and level hierarchies. As for limitations, very dense structures can cause occlusion which results in very incomplete point clouds leading to faulty inferred dependence or incorrect spatial transformations for movement. The current approach also requires multi-view perception and lacks feedback control to account for execution error due to hardware limitations such as forward momentum from the suction gripper. For future work, we aim to augment SCL with robust segmentation and point-cloud shape completion to reduce incorrect predictions. Another future objective is to extend SCL with multi-robot task allocation to use its existing ability of finding independent substructures to control multiple robots for faster execution. More avenues for exploration would include analysis of various graph networks and their application to structural planning and augmenting the dataset with objects having more diverse dimensions, leading to even more interesting structures. Lastly, we would like to extend SCL to more general multi-robot tasks and incorporate multi-agent specific considerations allowing for improved complex execution to accomplish more demanding target states, including non-monotone cases and those requiring in-place manipulation.

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

# 7 Appendix

## 7.1 Data Generation

The basics of data generation are 8 possible object primitives to construct structures. All objects may be present in the top layer in any valid orientation, but middle (or supporting) layers may only contain certain objects in certain orientations conducive to stable structures. For example, an upright pyramid would only provide a top surface of a single line and a cylinder that is not upright would be likely to roll and collapse the whole structure so these are not allowed. Overall, we initially generated 8000 scenes for training and 2000 scenes for evaluation. Our generator was also used to create upwards of 10000 on-the-fly target structures for the evaluation of our pipeline. A rough pseudo code for the structure generation algorithm is specified in Algorithm 2, but the generation heuristic will be open-sourced with the final manuscript. Note that there are some implementation details regarding the algorithm not specified here, like how the placement criteria has a metric involving the area of the top surface of objects, placement pose of objects is found by fitting a plane to sampled points from the top surface of the supporting objects in the lower level, validation of orientation for placement in layers is an exhaustive search, etc.

## 7.2 Model Architecture Details and Training

All graph neural networks were implemented using PyTorch Geometric (PyG) [38], and all conventional neural networks were implemented using PyTorch [39].

### 7.2.1 Positional Encoding

We utilized the positional encoding implementation specified in [32]. Specifically, the positional encodings $b_T^o$ for some object instance $o$ in the target scene are given by

$$b_T^o = \langle \sin(2^0 A x_o), \cos(2^0 A x_o), \ldots, \sin(2^L A x_o), \cos(2^L A x_o) \rangle \qquad (2)$$

where $x_o$ is the 3D position vector, the position associated with $o$ (which we get by calculating the centroid of the point cloud obtained from sampling the known default position, transformed to the target orientation), and the rows of $A$ are the outwards facing unit-norm vertices of a twice-tessellated icosahedron. We use no offset, a scale of 1, a min degree of 0, and a max degree of 5 to calculate $L$, which results in an encoding of size 511. For more details, please refer to [32].

### 7.2.2 PoinNet++ based segmentation

We used PyTorch Geometric's [38] example model for segmentation, as is, without significant changes. The input point clouds from each scene were downsampled to have 1024 points using random sampling, and training was done in a supervised manner using negative log-likelihood loss calculated from the output and the ground truth point identities extracted from the simulation. We trained on a set of 8000 scenes and validated performance on 2000 scenes before use.

### 7.2.3 PointNet++ based feature extraction

This utilized PyTorch Geometric's [38] implementation of PointNet++ with an MLP attached at the end to do classification on our set of 8 object primitives. Our MLP used had 3 layers. The first layer had an input size of 1024 and an output size of 512, the second layer had an input size of 512 and an output size of 256, and the final layer had an input size of 256 and an output size of 8. To obtain the object level features $w_T^{\{N\}}$ for each object in the target scene, we remove the final layer and take the 256-sized output to use as the object's latent features. The network was trained using a classification task for objects in over 800 scenes (each containing an average of 9 objects) and evaluated on objects in over 200 scenes before use.

### 7.2.4 GAT Graph Encoder

Our graph encoder $g_\Phi$ contains 2 graph attention convolution layers, each of which convolves around every node $i$ in the graph using its neighbor set $\mathcal{N}(i)$. The exact node feature update done by the

**Algorithm 2:** Generation-Algorithm($B, L^{\{K\}}, \mathcal{O}$). $B$ are the bounds for the scene, $L^{\{K\}}$ specifies the maximum number of objects on each level, $K$ specifies the number of levels, and $\mathcal{O}$ are all possible object instances that can be placed.

1  $\mathcal{O}_S \leftarrow$ ValidSubLevelObjects($\mathcal{O}$)
2  $c_0 \leftarrow 0$                              ▷ number of objects placed on level 0
3  $O \leftarrow \emptyset$                              ▷ objects placed
4  $A^{\{K\}} \leftarrow \emptyset$              ▷ $A_i$ contains all objects available for placement in level $i$
5  **while** $c_0 < L_0$ **do**
6  |  $o \leftarrow$ RandomlyPick($\mathcal{O}_S$)
7  |  **if** $o \in B$ and NotInCollision($o, O$) **then**
8  |  |  PlaceObject($o$)
9  |  |  $O \leftarrow O \cup \{o\}$
10 |  |  $A_0 \leftarrow A_0 \cup \{o\}$
11 |  |  $c_0 \leftarrow c_0 + 1$

12 $c^{\{N\}} \leftarrow 0$                        ▷ $c_i$ is the number of objects placed on level $i$
13 $i \leftarrow 1$
14 **while** $i \leqslant K$ **do**
15 |  **for** $(a, b) \in A_{i-1}$ **do**
16 |  |  $v \leftarrow$ ValidPlacementObjects($a, b, O, \mathcal{O}$)     ▷ obj instances that can be supported by $a, b$
17 |  |  $v \leftarrow$ CollisionFree($v, O$)          ▷ filters to give only collision free placements
18 |  |  **for** $o \in v$ **do**
19 |  |  |  PlaceOnObjects($o, a, b$)                  ▷ places $o$ to be supported by $a, b$
20 |  |  |  $A_i \leftarrow A_i \cup \{o\}$
21 |  |  |  $O \leftarrow O \cup \{o\}$
22 |  |  |  $c_i \leftarrow c_i + 1$
23 |  |  |  **if** $c_i < L_i$ **then**
24 |  |  |  |  **break**

25 |  |  **if** $c_i < L_i$ **then**
26 |  |  |  **break**

27 |  **for** $a \in A_{i-1}$ **do**
28 |  |  $v \leftarrow$ ValidPlacementObjects($a, O, \mathcal{O}$) ▷ object instances that can be supported by $a$
29 |  |  $v \leftarrow$ CollisionFree($v, O$)          ▷ filters to give only collision free placements
30 |  |  **for** $o \in v$ **do**
31 |  |  |  PlaceObject($o, a$)                        ▷ places $o$ to be supported by $a$
32 |  |  |  $A_i \leftarrow A_i \cup \{o\}$
33 |  |  |  $O \leftarrow O \cup \{o\}$
34 |  |  |  $c_i \leftarrow c_i + 1$
35 |  |  |  **if** $c_i < L_i$ **then**
36 |  |  |  |  **break**

37 |  |  **if** $c_i < L_i$ **then**
38 |  |  |  **break**

39 |  $i \leftarrow i + 1$
40 **return** O

convolutional layer is given by

$$n_i' = \alpha_{i,i}\Phi n_i + \sum_{j \in \mathcal{N}(i)} \alpha_{i,j}\Phi n_i \tag{3}$$

where $\Phi$ is the learnable parameter for the update mechanism and the attention coefficients $\alpha$, which quantifies the importance of neighboring nodes, is given by

$$\alpha_{i,j} = \frac{\exp(a^T \sigma(\Phi[n_i \,||\, n_j]))}{\sum_{k \in \mathcal{N}(i) \cup \mathcal{N}(j)} \exp(a^T \sigma(\Phi[n_i \,||\, n_k]))} \tag{4}$$

where $\sigma$ is the non-linear activation LeakyReLU with a slope parameter of 0.2. Each of the graph convolution layers had 16 attention heads with averaging used as the aggregation function. The first graph attention layer has an input size of 511 and an output size of 256 whereas the other graph attention layer has an input size of 256 and an output size of 128. Training was done in a supervised manner with

### 7.2.5 MLP Edge Decoder

The edge decoder $h_\Psi$ can be queried with a pair of high-level features $z_i, z_j$, resulting from the graph network encoder, and will decode them into the structural relationship between them. This relationship of dependence is asymmetric, and the decoder is used to query every ordered pair of nodes in the graph to obtain the respective dependence probabilities $\rho^{\{N \times N\}}$ where $\rho_{i,j} = h([z_i || z_j]; \Psi)$. $h_\Psi$ has 2 fully connected layers and uses LeakyReLU as its non-linear activation function after the first layer only. The first layer has an input size of $2 \cdot 128 = 256$ and an output size of 128, whereas the second layer has an input size of 128 and an output size of 1. We apply SoftMax to get the associated probabilities for training with binary cross-entropy loss. The reason for the input layer for $h_\Psi$ being twice the output size for $g_\Phi$ is because finding the existence probability for the edge $(i, j)$ involves concatenating the high-level node features $z_i, z_j$ before inputting them into the edge decoder. The graph encoder $g_\Phi$ and edge decoder $h_\Psi$ were trained together in a supervised manner to minimize the binary cross entropy loss between the adjacency matrices of the predicted and ground truth dependency graphs, which, in turn, was obtained from simulation information (specifically, the $y$-components of the contact force vectors on each pair of objects).

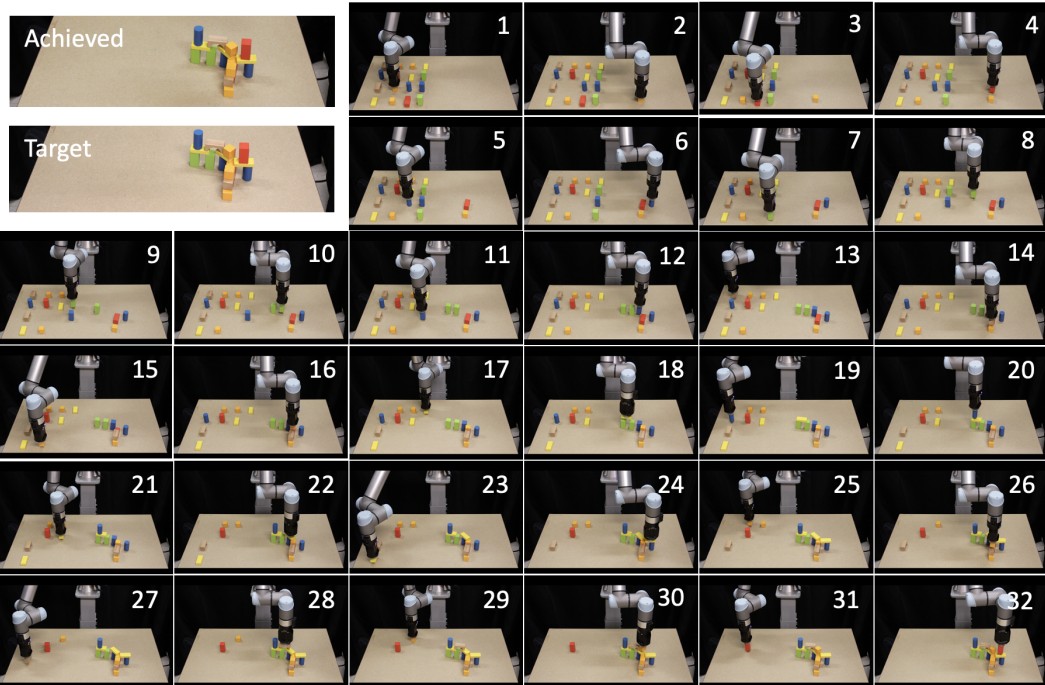

Figure 4: Our approach performing progressive pick-and-place (with all steps shown) actions based on its multi-level rearrangement plan to achieve (top left) a target arrangement (middle left).

