# OpenReview forum: "Structural Concept Learning via Graph Attention for Multi-Level Rearrangement Planning"
_robot-learning.org/CoRL/2023/Conference — CoRL 2023 Poster_

### Official Review · Reviewer_mpK3 · 2023-07-12

**Confidence:** 3
**Originality:** Good
**Technical Quality:** Very Good
**Clarity Of Presentation:** Very Good
**Impact:** 3

**Recommendation:**

Weak Accept: I recommend accepting the paper, but will not argue for my recommendation if the majority of other reviewers have a different opinion.

**Review:**

Strengths of the paper:

1: Building the stack with a meaningful structure is interesting.

2: The real-world demo with a long-horizon sequence of actions is cool.

3: This approach is trained purely in simulation and can generalize to real-world without any fine-tuning.

Weakness of the paper:

1: The assumptions of the paper are strong. This paper assumes three cameras to get a calibrated point cloud and then leverages this calibrated point cloud to calculate the spatial transformations.

2: The assumption of known spatial transformations makes the task much easier. Thus, even if the SCL performs great, one of the baselines (classical random) also performs more than a 90% success rate and completion rate.


Minor points:

1: Figure 2 is hard to follow and understand.

2: multi-head instead of multi-heads.

**Quality Of The Limitations Section:**

Additional details required

**Questions For Rebuttal:**

1: The authors should discuss more how to learn the spatial transformations as an end-to-end learning.


**Robotics Focus:**

Sufficient demonstration on hardware

**Summary Of Paper:**

This paper proposes an approach based on a graph attention network for multi-level rearrangement tasks. This approach can reason about the dependency between objects at the same or different levels. With the reasoning of the dependency, this approach can sequentially pick and place objects to build a goal structure. Furthermore, based on the training in simulation, the framework can generalize to real-world scenarios.

**Summary Of Recommendation:**

This paper proposes an approach based on a graph attention network to solve multi-level rearrangement tasks. This task is interesting for the community and the real-world demo with long-horizon executions is good. The assumptions are strong but if the authors discuss more about how to solve this in future work, this will be a useful paper for CoRL. Therefore, I recommend "weak accept".

---

> ### Author Response · Authors · 2023-08-14
> **A gentle and kind reminder for author-reviewer discussion**
>
> Dear reviewer “mpK3”, thank you for your detailed feedback in your reviews! Since the revision period will end tomorrow, we would appreciate it if we could receive your response to our revisions and rebuttal. We will be happy to address any further concerns and modify our paper accordingly.

---

### Official Review · Reviewer_fLRU · 2023-07-19

**Confidence:** 5
**Originality:** Very Good
**Technical Quality:** Very Good
**Clarity Of Presentation:** Excellent
**Impact:** 3

**Recommendation:**

Weak Accept: I recommend accepting the paper, but will not argue for my recommendation if the majority of other reviewers have a different opinion.

**Review:**

Strengths:

1. The paper was a pleasure to read, offering a well-articulated and lucid description of the implementation details.
2. The approach of employing a graph architecture to model and infer the relationships between blocks is both innovative and logical.
3. The real-world experiment outcomes were both promising and engaging to observe, which points to a good system design.

Limitations:

1. The enhancements demonstrated by the proposed method appear marginal when contrasted with the traditional random baseline in Table 1. It shows only a 5% improvement in task success rate, which raises questions about the necessity and efficacy of the proposed architecture.
2. The proposed method hinges on an assumption that two rotation axes of an object must remain in their initial state, as the robot cannot perform reorientation along these two axes (primarily x and y axes). This assumption substantially constrains the task's state space. It would be beneficial for the authors to explicate this assumption within the problem formulation.
3. The overall framework appears to function as an open-loop controller, which may lack the capability to rectify failures in a closed-loop manner. Is there a re-planning mechanism designed to handle unexpected failures?
4. Why not employ the simulation directly as a model and use Model Predictive Control (MPC) to generate viable task plans? Given that the task is kinematic pick-and-place without dynamic movements, what are the primary challenges for achieving sim-to-real generalization in this context?
5. How can the proposed method be applied to more significant and impactful manipulation domains?
6. What is the distribution of objects in the initial state? Does the model have the capacity to dismantle an existing structure and construct a new one?

**Quality Of The Limitations Section:**

Limitations are addressed clearly

**Questions For Rebuttal:**

1. Could you justify the proposed architecture given the 5% improvement over the baseline?
2. Can you clarify the impact of the fixed rotation axes assumption on the method's robustness?
3. How does your system manage unexpected failures, considering it operates as an open-loop controller?




**Robotics Focus:**

Sufficient demonstration on hardware

**Summary Of Paper:**

This paper focuses on the rearrangement planning task, which involves the reorganization of objects in a given environment to bring them into a goal state. The paper particularly explores a scenario of constructing a structure with known object primitives. The proposed method takes a multi-view RGB-D observation of the target structure and constructs a dependency graph using graph attention networks, which captures the geometrical dependency among different objects, identifying independent substructures for parallelization. Then, a structure planner serializes task executions based on the current observations of the scene. A low-level controller further executes the tasks via robot control. The results demonstrate superior performance and generalizability compared to other classical and model-based planners.

**Summary Of Recommendation:**

The paper provides valuable insights into rearrangement planning, proposing an innovative method that showcases potential scalability and generalizability. Despite modest improvements over the baseline, its contributions to the dataset and procedure generation make it a noteworthy addition to the field.

---

### Official Review · Reviewer_Ej1j · 2023-07-27

**Confidence:** 4
**Originality:** Good
**Technical Quality:** Very Good
**Clarity Of Presentation:** Very Good
**Impact:** 4

**Recommendation:**

Weak Accept: I recommend accepting the paper, but will not argue for my recommendation if the majority of other reviewers have a different opinion.

**Review:**

Strengths:
- The authors clearly define the problem they want to solve.
- The usages of graph node encoder and MLP edge decoder are very novel. The graph node encoder extracts the high-level features for each of the objects within the scene. The MLP edge decoder recovers the inter-object specific dependency information.
- The planning algorithm is well-explained and makes sense.
- The experiments are quite thought-out. They show the model performs well on challenging problems defined by target scenes containing different unseen structures with a variety of objects (up to 25 objects) and level hierarchies (up to 5 levels).
- The work performs experiments on real world robots. Thus, it shows the feasibility of the approach for practical application.

Weaknesses:
- The work claims the approach allows for task parallelization over multiple manipulators but fails to show any result to support that.
- The work mentions 5 sets of experiments, but it only shows the results for 4 of them.
- Do all the compared methods have the same run time? If not, the authors should include the total run time for each compared method. Since both classical iterative baseline and classical random baseline have an iterative approach for solving the planning problem and the time they run will affect their performance.
- In the limitation section, the authors mention dense structures can cause occlusion and in future work, the authors plan to augment SCL with robust segmentation, point cloud shape completion, and matching techniques. However, most of these are gotten from
other works (such as PointNet++ and TEASER). So, these limitations seem to be not from the proposed work but from the other works. The discussion of limitations is a bit lacking.
-  There are a few typos within the paper such as the figure 2 caption.




**Quality Of The Limitations Section:**

Limitations are addressed clearly

**Questions For Rebuttal:**

- What is the success threshold for the completion rate? What is the exact size of the objects, or their relative size compared to the environments? Without knowing those, the position errors do not provide much information.
- Why are the classical baselines given a budget of 2.N since it’s stated that 2.N is too little for an iterative approach? Is N the number of nodes or iterations? From the previous section, N is the number of nodes, then how does it affect the planning?
- In the data generation, the work mentions that after the first layer, there is a 2-step process for each layer in which they attempt to place an object by 2 objects in previous layers and then 1 object in the previous layer. Is it possible to have an object to be supported by more than 2 objects in previous layers? Could a random number (1-4) be chosen for this procedure? Would that generate more diverse training datasets? From the experiments, we know that there could be up to 25 objects in a scene.

**Robotics Focus:**

Sufficient demonstration on hardware

**Summary Of Paper:**

The paper proposed Structured Concept Learning, a graph attention network-based approach for multi-level rearrangement planning. Particularly, their approach takes a multi-view RGB-D observation of the target structure and constructs a dependency graph using graph attention networks. First, the model segments out object point clouds, perform pose alignment, and establish object correspondences to different scenes. The object level embeddings and positional embeddings create a graph that would be used by the graph node encoder to give higher-level node features. Then the edge decoder uses this information to create a dependency graph from which the planner outputs a valid sequence for robot rearrangement. The work conducts 4 types of experiments to show their performance, flexibility, and efficiency. They also performed a set of real-world experiments to demonstrate their generalization.

**Summary Of Recommendation:**

The paper proposed a novel idea in the usage of node encoder and edge decoder to extract extracts the high-level features for each of the objects and recovers the inter-object specific dependency information for planning. Most of the experiments are well conducted, show the strength, and provide insight for the model. However, there seems to be one missing experiment to show the execution time and parallelization over multiple manipulators. There are also some missing information and explanation.

---

### Official Review · Reviewer_hfsk · 2023-07-29

**Confidence:** 3
**Originality:** Very Good
**Technical Quality:** Very Good
**Clarity Of Presentation:** Very Good
**Impact:** 3

**Recommendation:**

Weak Reject: I recommend rejecting the paper, but will not argue for my recommendation if the majority of other reviewers have a different opinion.

**Review:**

The paper presents an innovative approach for multi-level object rearrangement planning using graph attention networks. The proposed SCL method demonstrates promising results in handling complex scenes with structural dependencies, and its generalization to the real world is a significant contribution. The paper is generally well-written and provides a clear description of the method's architecture and implementation details. The use of a self-generated simulation dataset and comparison against baselines strengthens the paper's empirical evaluation.

The main strength of the paper lies in its novel combination of graph attention networks for multi-level planning, which addresses an important problem in robotic manipulation. The method's performance, flexibility, and efficiency make it a valuable contribution to the field.

However, some weaknesses should be addressed. First, the paper relies on a synthetic dataset, and it would be beneficial to evaluate the method on real-world data to assess its robustness and practical applicability. Second, the complexity and efficiency of the approach should be discussed further, as the method may be computationally demanding for large scenes or multiple manipulators.Moreover, the paper lacks a clear explanation of how GAT's attention mechanism is relevant to hierarchical planning. A more detailed discussion of the advantages of using GAT over other standard Graph Neural Networks (GNNs) like Graph Convolutional Networks (GCN) or GraphSAGE would strengthen the paper's arguments.

**Quality Of The Limitations Section:**

Additional details required

**Questions For Rebuttal:**

I fail to see the benefits of using GAT for multi-level rearrangement planning. The paper claims that a GAT-based model with multi-headed attentions outperformed vanilla GCNs. Can you provide theoretical or empirical evidence to support this claim?

There are other standard GNN structures like GraphSAGE. Why was GAT chosen over these alternatives? Is there a specific advantage of GAT in the context of this problem?

What is the depth of the GAT model used in this work? DeepGCNs have shown good results in point cloud segmentation by going deep. Though here, it is used as an encoder, can you provide insights into the choice of the GAT depth?

The process of obtaining edge embeddings based on node embeddings using an MLP decoder may lead to loss of topology information. For instance, nodes zi1, zi2 with the same embedding may have different relationships. Can you clarify the benefits of obtaining a new adjacency matrix in this manner? Why not directly use GAT's weight matrix as the new adjacency matrix? It seems like you are using learned nodes to infer the learned adjacency matrix, and I am unsure of the advantages of using an MLP decoder over GAT's adjacency matrix.

Overall, I did not observe the advantages of using GAT in Rearrangement Planning based on the presented results. Can you provide more insights or explanations to highlight the benefits of GAT in this context?

**Robotics Focus:**

Relevant but unlikely to deploy to hardware in near future

**Summary Of Paper:**

The paper proposes a method called Structural Concept Learning (SCL) for multi-level object rearrangement planning in robotic manipulation tasks. SCL leverages graph attention networks to generate a hierarchical plan for scenes with structural dependency hierarchies. It uses a self-generated simulation dataset with intuitive structures and can handle scenes with an arbitrary number of objects and complex structures. The method infers independent substructures to enable task parallelization over multiple manipulators and generalizes well to the real world. The proposed approach is compared against classical and model-based baselines, demonstrating better performance, flexibility, and efficiency.

**Summary Of Recommendation:**

The paper proposes SCL, a novel multi-level object rearrangement planning method using graph attention networks (GAT). The approach demonstrates promise in handling complex scenes with structural dependencies, making it valuable for robotic manipulation tasks. However, several weaknesses need addressing. Firstly, evaluating the method on real-world data is crucial to assess its robustness and practical applicability. Secondly, discussing the complexity and efficiency of the approach, particularly for larger scenes or multiple manipulators, would enhance its practicality. Additionally, the paper lacks a clear explanation of how GAT's attention mechanism benefits hierarchical planning. Comparing GAT's performance against other standard GNNs, like GCNs or GraphSAGE, would bolster the paper's claims. The authors are encouraged to address these concerns in the revision to strengthen the paper's impact and significance in the field.

---

> ### Author Response · Authors · 2023-08-14
> **A gentle and kind reminder for author-reviewer discussion**
>
> Dear reviewer “hfsk”, thank you for your detailed feedback in your reviews! Since the revision period will end tomorrow, we would appreciate it if we could receive your response to our revisions and rebuttal. We will be happy to address any further concerns and modify our paper accordingly.

---

### Author Response · Authors · 2023-08-06
**Summary for area chair and reviewers**

We would like to thank all the reviewers for all their work in reviewing our project and providing us with detailed comments and constructive criticism to help us make it better. We have posted responses to each of the reviews and made any associated changes to the manuscript, with additions and changes highlighted in red. The revised manuscript (pdf) is attached to all the rebuttal responses. In addition to our responses, we would also like to highlight the following:
- We selected GAT because it has a higher F1 score than GCN and a similar F1 score to GraphSage. However, we did not choose GraphSage, which is more commonly used for larger graphs with thousands of nodes. It is important to note that our paper's goal is to use graph neural networks for structural planning rather than compare them to one another.
- We highlight that our method is scalable as its computational times remain similar for a different number of objects.
- We have revised our limitations description in Section 6 and appropriately motivated our future work to address them. The revised manuscript is attached to all rebuttal responses.
- We have fixed minor typo-related issues pointed out by the reviewers.
- We will be releasing the dataset, the associated heuristic and code for generating it, and the implementation of our approach.

---

### Decision · Program_Chairs · 2023-08-30

**Decision:**

Accept (Poster)

**Comment:**

The paper presents a novel approach to the rearrangement planning task, focusing on the reorganization of objects in an environment to achieve a goal state. The method leverages multi-view RGB-D observations and graph attention networks to capture geometrical dependencies between objects, enabling the identification of independent substructures for parallelization. A structure planner and low-level controller execute tasks, showcasing promising real-world performance and generalizability. The paper introduces an innovative approach using graph attention networks to model object relationships, enhancing the understanding of the dependencies between blocks in the rearrangement task. The real-world experiments demonstrate the efficacy of the proposed approach, showcasing its potential for scalability and generalizability.
Concerns were raised about the modest improvement (5%) over the traditional random baseline.  The authors justified this by highlighting the budget constraints and the ability of their approach to execute tasks within N steps. There were queries regarding the assumption that two rotation axes of an object remain fixed.The approach appears open-loop, lacking a mechanism to rectify failures. Authors acknowledge this and leave handling unexpected failures for future work, suggesting that dependency graph errors can be recovered from. The question of employing Model Predictive Control (MPC) for generating task plans directly from simulation was raised and in the rebuttal stage, the authors highlight the computational cost of MPC and emphasize their approach's scalability.
The paper introduces some contribution to the field of rearrangement planning, leveraging graph attention networks for modeling dependencies and achieving promising real-world results. While some concerns were raised regarding the improvements over baselines and assumptions, the authors' responses addressed most of the queries satisfactorily. The paper's technical quality, clarity, and innovation were generally acknowledged.